# Position: Hallucinations Undermine Trust; Metacognition is a Way Forward

**Gal Yona** [1]  **Mor Geva** [2]  **Yossi Matias** [1]

## Abstract

Despite significant strides in factual reliability, errors—often termed hallucinations— remain a major concern for generative AI, especially as LLMs are increasingly expected to be helpful in more complex or nuanced setups. Yet even in the simplest setting—factoid question-answering with clear ground truth — frontier models without external tools continue to hallucinate. We argue that most factuality gains in this domain have come from *expanding the model's knowledge boundary* (encoding more facts) rather than *improving awareness of that boundary* (distinguishing known from unknown). We conjecture that the latter is inherently difficult: models may lack the discriminative power to perfectly separate truths from errors, creating an unavoidable tradeoff between eliminating hallucinations and preserving utility. This tradeoff dissolves under a different framing. If we understand hallucinations as *confident* errors—incorrect information delivered without appropriate qualification—a third path emerges beyond the answer-or-abstain dichotomy: expressing uncertainty. We propose *faithful uncertainty*: aligning linguistic uncertainty with intrinsic uncertainty. This is one facet of *metacognition*—the ability to be aware of one's own uncertainty and to act on it. We conclude by highlighting open problems for progress towards this objective.

## 1. Introduction

Despite significant strides in factual reliability (Tian et al., 2023a; Wei et al., 2024b; Grattafiori et al., 2024; Cheng et al., 2025), errors—often termed "hallucinations"—remain a major concern for generative AI, especially as large language models (LLMs) are increasingly expected to be helpful in more complex or nuanced setups. These factually

incorrect generations are often delivered with an authoritative tone, risking undermining user trust and spreading misinformation (Ji et al., 2023; Zhang et al., 2025b; Steyvers et al., 2025).

In this paper, we focus on a simple setting where frontier models still hallucinate: factoid question-answering with clear ground truth (setting aside long-form generation and cases of genuine ambiguity or contested claims). For models without access to external tools, we argue that **most factuality gains in this domain have come from *expanding the model's knowledge boundary* (encoding more facts) rather than *improving awareness of that boundary* (distinguishing known from unknown)**. We conjecture that this asymmetry arises because while expanding the knowledge boundary is often achievable through scale, data, and improved training recipes, **models may fundamentally lack the discriminative power to perfectly separate truths from errors**.

Although research on uncertainty quantification has shown that well-calibrated confidence signals can be extracted from LLMs (Kadavath et al., 2022; Lin et al., 2022; Tian et al., 2023b; Nakkiran et al., 2025), *calibration* (confidence scores matching the probability of correctness) does not guarantee *discrimination* (confidence scores that can sharply distinguish correct from incorrect answers). **Under the traditional view that treats hallucination as synonymous with error, limited discrimination creates an inherent tradeoff between eliminating hallucinations and preserving utility:** to guarantee zero hallucinations, a model must abstain whenever uncertain, suppressing valid information along with errors. **In practice, model providers are often unwilling to pay this "utility tax," resulting in models that prioritize answering and still hallucinate** (Fig. 1, left).

This perspective ties together many recent empirical observations. The poor generalization of truthfulness probes (Levinstein & Herrmann, 2023; Orgad et al., 2025; Sky et al., 2024; Marks & Tegmark) and the existence of confident hallucinations (Simhi et al., 2025; Wang et al., 2025b; Taubenfeld et al., 2025) both demonstrate the limits of model internals for veracity prediction. The failure of advanced alignment techniques, such as training models to "confess" errors (Joglekar et al., 2025), to mitigate hallucinations confirms

[1]Google Research [2]Tel Aviv University. Correspondence to: Gal Yona <gal.yona@gmail.com>.

*Proceedings of the 43^{rd} International Conference on Machine Learning*, Seoul, South Korea. PMLR 306, 2026. Copyright 2026 by the author(s).

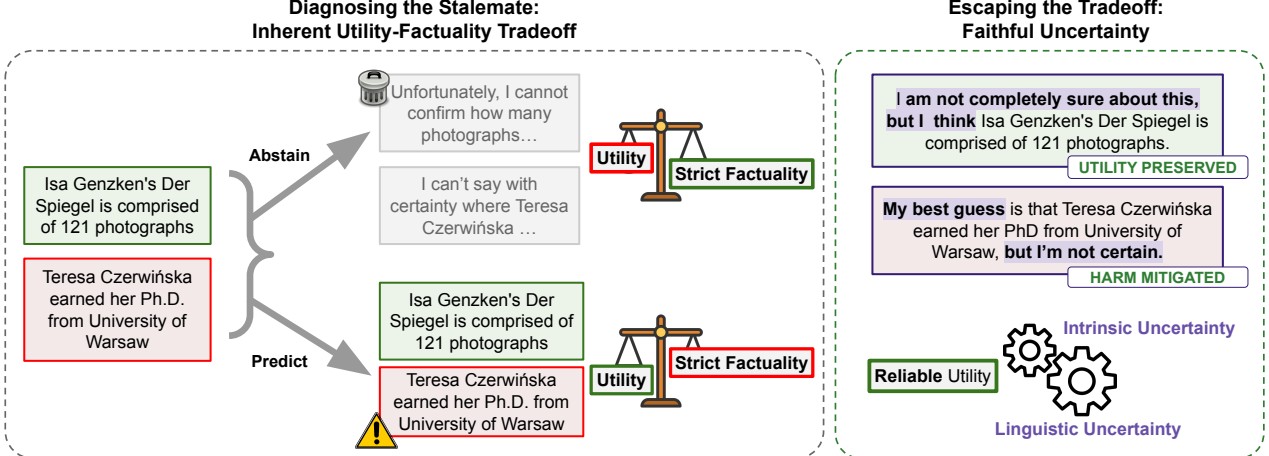

*Figure 1.* **Escaping the Utility-Factuality Trade-off. (Left)** We conjecture that models lack the discriminative power to perfectly separate truths (green) from errors (red). Under the traditional view that any error constitutes a hallucination, this creates a dilemma: the model must either *abstain* (top), paying a "utility tax" by suppressing valid information (🗑); or *predict* (bottom), risking confident errors that erode trust (⚠). **(Right)** We propose reframing hallucination as *confident* error, which reveals a third path: *faithful expression of uncertainty*. By aligning linguistic output with intrinsic confidence, the model retains valid information as appropriately hedged estimates while rendering errors less harmful.

that such deficits persist under strong supervision. Together, these provide evidence for the existence of a discrimination gap. Finally, the surprising finding that extended reasoning *increases* hallucinations (Jaech et al., 2024; Yao et al., 2025; Li & Ng, 2025) and degrades abstention (Kirichenko et al., 2025; Zhang et al., 2025a) reflects how current training incentivizes models to favor utility in the presence of these tradeoffs.

**However, the apparent tradeoff between trust and utility dissolves under a different framing.** If we understand hallucinations not as any error, but as **confident errors**—incorrect information delivered without appropriate qualification—**a third path emerges: expressing uncertainty.** An error communicated with appropriate hedging is not a hallucination; it is a hypothesis offered for consideration. The model need not choose between utility and trust; it can preserve both by honestly communicating its uncertainty.

Naturally, simply hedging more is not the answer. A model that hedges uniformly provides no signal; the hedge can technically be calibrated to error rates while being completely uninformative at the instance level. **What is needed is *faithful* uncertainty: hedging that reflects the model's actual internal state for each specific answer.** This builds on the notion proposed by Yona et al. (2024) and later Ghafouri et al. (2024), which requires aligning the model's *linguistic uncertainty* (what it says) with its *intrinsic uncertainty* (what it "believes"). Faithful uncertainty is one facet of what we call *metacognition*—the ability to be aware of one's own un-

certainty and to act on that awareness. In direct interaction, acting on uncertainty means communicating it honestly; as we discuss later, for agentic systems it means using uncertainty to guide tool use.

**Faithful uncertainty ensures the model's response provides an honest signal of its internal state, with clear behavioral semantics**: "I am confident" means the model would likely give the same answer if asked again; "I am uncertain," suggests it is likely to give a conflicting answer. This is information users can act on, regardless of whether the model is ultimately correct. Crucially, **faithful uncertainty is feasible in principle**: it depends only on the model's internal states, not on solving the difficult problem of knowing when those states correspond to truth. A model may not perfectly know when it is wrong, but it *can* know when it is uncertain.

**This framing acknowledges that trust can be built on imperfect knowledge, provided uncertainty is communicated honestly**—just as we trust a doctor not for omniscience, but for reliably distinguishing diagnoses from hypotheses. It also becomes increasingly urgent as models grow more capable: **as outputs become more sophisticated, they become harder for users to verify independently, making honest communication of uncertainty a safety requirement**.

While faithful uncertainty addresses cases where the model is intrinsically uncertain, **the remaining errors—where the model is genuinely confident but wrong—are "honest mistakes,"** addressable only through continued knowl-

edge expansion. This highlights how the two efforts are complementary: knowledge expansion pushes the knowledge boundary further; faithful uncertainty communicates whatever boundary remains.

Metacognition has a second facet beyond expression: for agentic systems, it becomes the control layer. The shift to agentic AI systems **effectively expands the knowledge boundary**—the model can retrieve information it does not have encoded. On the surface, this might suggest that awareness of uncertainty becomes redundant: *why know what you don't know if you can simply look everything up?* **But awareness of uncertainty is precisely what enables effective tool use.** Without it, a model cannot determine when to invoke a tool (leading to inefficient overuse or dangerous underuse), nor can it appropriately weigh retrieved information against its own beliefs when conflicts arise. Faithful uncertainty is thus not circumvented by tools, but rather becomes the control layer that governs them (Rabanser et al., 2026). Yet modern search agents lack such awareness, leading to inefficient tool overuse (Qian et al., 2025; Lin et al., 2025). Instilling metacognition thus addresses not only parametric reliability, but provides the foundation for robust agentic behavior.

While LLMs are currently poor at faithfully conveying their uncertainty, we believe this problem offers tangible headroom. Recent work demonstrates promising directions through metacognitive prompting (Liu et al., 2025), fine-tuning (Eikema et al., 2025), and model internals (Ji et al., 2025). Encouraging results also include the observation that reasoning models better express their confidence (Yoon et al., 2025; Podolak & Verma, 2025) and the success of intrinsic signals as rewards in reinforcement learning (Prabhudesai et al., 2025; Wang et al., 2025a; Li et al., 2025b).

> ### Summary of Contributions
>
> - We argue that most factuality gains to date in factual question-answering with clear ground truth have come from expanding the model's knowledge boundary rather than improving awareness of that boundary, and conjecture that the latter may be fundamentally difficult due to limited discriminative power.
>
> - We propose reframing hallucination as *confident* error rather than mere error. This reveals a third path beyond the answer-or-abstain dichotomy: expressing uncertainty. Faithful uncertainty—the metacognitive capability of aligning linguistic uncertainty with intrinsic uncertainty—directly mitigates hallucinations (as reframed) while preserving utility (Figure 1, right). Unlike calibration, which is an aggregate property, faithfulness pro-

> vides an instance-level guarantee: each hedge reflects that specific answer's internal state. This approach complements continued efforts in knowledge expansion, which addresses the remaining "honest mistakes" where models are confident but wrong.
>
> - We introduce concrete recommendations for evaluating hallucination mitigation techniques, including prioritizing discriminative measures over calibration and holistically quantifying the utility cost of interventions. For researchers interested in faithful uncertainty and metacognition, we sketch key open problems as entry points.

**Organization.** We begin with background on definitions, metrics, and mitigation strategies (§2), then present our analysis of the challenges facing strict factuality (§3). Next, we propose the objective of faithful uncertainty (§4), arguing for its feasibility and favorable utility-reliability properties, and discuss its role in agentic systems (§5). Finally, we outline concrete recommendations for researchers (§6), address alternative viewpoints (§7), and conclude (§8).

## 2. Background

### 2.1. Problem Scope

**Extrinsic Hallucinations in Parametric Models.** We distinguish between two modes of deployment. Parametric LLMs rely on their own parameters (Petroni et al., 2019; Roberts et al., 2020), while Tool-Augmented LLMs interact with external sources, such as search engines or APIs, to retrieve information at inference time (Lewis et al., 2020; Nakano et al., 2021; Yao et al., 2022; Schick et al., 2023). We primarily focus on the former and discuss implications for tool-use in §5. Within this scope, we specifically target extrinsic hallucinations – generations that are factually incorrect with respect to real-world knowledge (Ji et al., 2023; Huang et al., 2025) – as opposed to intrinsic hallucinations (contradicting source text) or reasoning errors.

**The Challenge of Tail Knowledge.** Standard hallucination mitigation evaluations often center on common misconceptions (Lin et al., 2021) or head knowledge (Joshi et al., 2017; Kwiatkowski et al., 2019), potentially masking the severity of hallucinations in the regime of sparse data (Kandpal et al., 2023; Mallen et al., 2023). To probe the true boundaries of reliability, we focus on tasks that require "long tail" knowledge, using benchmarks (Wei et al., 2024a; Haas et al., 2025; Mallen et al., 2023; Jackson et al., 2025) that ask explicit and simple questions about facts regarding very rare entities (e.g., *"On which U.S. TV station did the Canadian reality series To Serve and Protect debut?"*).

## 2.2. Measuring Reliability

**The Utility-Factuality Trade-off.** Evaluating hallucinations requires nuance, because a model can trivially achieve zero hallucinations by refusing to answer any question where it is not certain. While achieving zero hallucinations, this strategy renders the model practically useless. Robust evaluation must therefore track both *accuracy* (overall correctness) and *attempted accuracy* (correctness on the subset for which an answer was attempted). Since in practice the knowledge captured in the model is bounded, the ideal behavior is to maximize both accuracy and attempted accuracy, using summary metrics like F1 (Wei et al., 2024a) or Omniscience Index (Jackson et al., 2025).

**Calibration vs. Discrimination.** Central to our argument is the distinction between two dimensions of uncertainty quantification. *Calibration* measures the alignment between confidence scores and empirical accuracy; scores are perfectly calibrated if, among all predictions assigned confidence $p$, exactly $p\%$ are correct. In contrast, *discrimination* measures the ability to distinguish correct from incorrect answers based on confidence.[1] Crucially, calibration does not imply discrimination. A confidence score that assigns a static confidence of 0.6 to every answer (and is correct 60% of the time) is perfectly calibrated yet has zero discriminative power. As we argue in §3, eliminating hallucinations in practice requires good discrimination, not just calibration.

## 2.3. Existing Mitigation Strategies

Research on mitigating hallucinations in parametric LLMs has generally followed two streams. Training-time interventions include careful data filtering and regularization (Gekhman et al., 2024; Kaplan et al., 2026), penalizing non-factual outputs (Ouyang et al., 2022; Tian et al., 2023a), using rewards based on collaborative games (Eisenstein et al., 2025) and mitigating overconfidence via linguistic calibration (Mielke et al., 2022; Yang et al., 2024; Stengel-Eskin et al., 2024). Inference-time interventions focus on steering the model toward factual generations without altering weights, using custom decoding strategies (Chuang et al.; Shi et al., 2024), relying on internal signals (Li et al., 2023b; Yu et al., 2024) or self-verification (Cohen et al., 2023; Dhuliawala et al., 2024).

## 3. Why Hallucinations Persist

Having established the background and metrics, we now turn to analyze the state of the field. In this section, we argue that the objective of fully eliminating hallucinations

faces fundamental challenges. We ground this position in theoretical limits, information bottlenecks, and empirical evidence from state-of-the-art LLMs.

## 3.1. The Theoretical Ceiling

Previous work argued that extrinsic hallucinations are a structural inevitability of auto-regressive text generation. Banerjee et al. (2025) and Xu et al. (2024) utilized the Halting Problem and diagonalization arguments to prove that no computable model can universally verify truth or learn all ground-truth functions. Kalai & Vempala (2024) showed calibrated models are mathematically bound to hallucinate when generating facts whose truth value cannot be inferred from other facts, while Kalavasis et al. (2025) established a formal trade-off between consistency and breadth—proving that reducing hallucination rates below a critical threshold necessitates a drastic reduction in output diversity, inevitably forcing the model into mode-collapse.

## 3.2. The Discriminative Gap

Previous work has largely focused on the positive results in eliciting well-calibrated confidence signals from LLMs (Kadavath et al., 2022; Tian et al., 2023b; Zhao et al., 2024; Nakkiran et al., 2025). While this demonstrates that LLMs can accurately aggregate uncertainty, there is a critical distinction between knowing the average error rate (calibration) and knowing which specific instances are errors (discrimination). We conjecture that the practical failure of standard mitigation techniques stems precisely from this deficit—a fundamental lack of discriminative power. Indeed, a weak correlation between these two metrics has been demonstrated in practice for many confidence scores (Savage et al., 2025; Taubenfeld et al., 2025; Tao et al., 2025). We further illustrate this point in Figure 2, showing a practical scenario in which using a well-calibrated confidence signal (left) to eliminate hallucinations results in major trade-offs with utility (right). E.g., to reduce the error rate from 25% to a target of 5%, the model must discard over 50% of its valid answers.

To empirically quantify the discrimination gap, we review AUROC values from the literature for the task of separating correct from incorrect answers using a model's confidence signal (AUROC = 1.0 is perfect; 0.5 is random). Across methods, models, and tasks, AUROC clusters in the 0.70–0.85 range for realistic factual QA tasks (Farquhar et al., 2024; Savage et al., 2025; Kang et al., 2025). Concretely, Farquhar et al. (2024) report an average AUROC of 0.79 across 30 model×task combinations using semantic entropy; Savage et al. (2025) find GPT-4 tops out at 0.79 in medical QA; and Kang et al. (2025) find GPT-4o-mini reaches only 0.68–0.72 on biography generation, which resembles our tail facts setting. Crucially, this range is insufficient to

---

[1] A similar distinction has been drawn in the metacognition literature in cognitive science, where the analogous concept, *resolution*, has been argued to be more diagnostic of metacognitive accuracy than calibration (Nelson, 1984; Fleming & Lau, 2014).

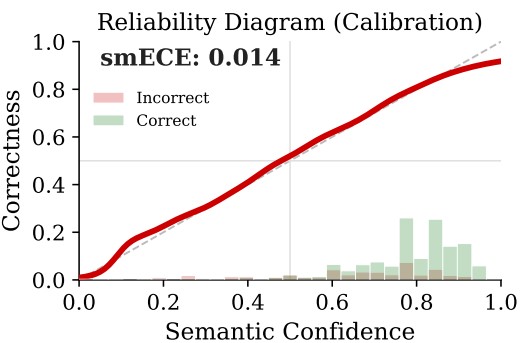
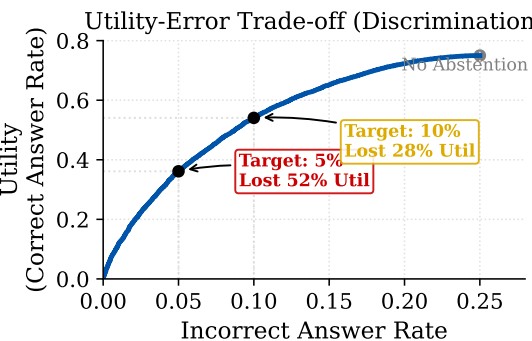

*Figure 2.* **Calibration vs Discrimination**. **Left**: We simulate data (see App. A) to match the reliability diagram in Nakkiran et al. (2025) (Figure 3). The model has a 25% base error rate and achieves strong calibration (measured w. SmoothECE; Blasiok & Nakkiran), indicated by the curve hugging the diagonal, though the underlying histogram shows that correct and incorrect answers often share similar confidence scores. **Right**: The Utility-Error Trade-off curve illustrates the cost of fully eliminating hallucinations. Despite good calibration, reducing the hallucination rate from 25% (No Abstention) to a strict target of 5% requires discarding 52% of valid answers (Utility). This visualizes the utility tax: without very strong discrimination, eliminating hallucinations requires suppressing a massive volume of correct information.

escape the utility tax. The simulation underlying Figure 2 has AUROC = 0.71, consistent with the literature average; at this level, reducing the error rate from 25% to a 5% target requires discarding 52% of valid answers. Even at the 0.85 ceiling, the tax remains ~28%. The tax only becomes negligible (<5%) at AUROC ≥ 0.95, which is far above any method currently reported for knowledge-intensive tasks.

### 3.3. Corroborating Anomalies

Recent anomalies in model development corroborate these theoretical constraints, mapping directly to the conjectured discrimination gap. First, the poor generalization of truthfulness probes (Levinstein & Herrmann, 2023; Orgad et al., 2025; Sky et al., 2024; Marks & Tegmark) and the demonstrated existence of "confident hallucinations" – factual errors with high intrinsic confidence (Simhi et al., 2025; Wang et al., 2025b; Taubenfeld et al., 2025) – demonstrate that in practice, the information required to robustly distinguish correct from incorrect answers is often absent even from the model's latent states. This also explains the failure of advanced supervision in practically mitigating hallucinations. Joglekar et al. (2025) demonstrate that while models can be aligned to "confess" to intentional safety violations, this capability fails to transfer to hallucinations. This divergence indicates that unlike safety issues, hallucinations are not merely behavioral bugs but stem from the discrimination gap; the model cannot be aligned to report errors it cannot internally represent. Finally, in the absence of reliable discrimination, optimizing for utility actively exacerbates hallucination. Recent work indicates that "thinking" often increases hallucination rates (Jaech et al., 2024; Yao et al., 2025; Li & Ng, 2025) and degrades abstention (Kirichenko et al., 2025), with increased performance gaps between an-

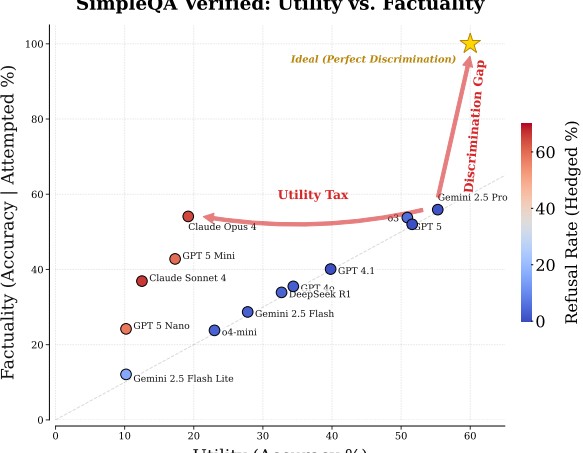

*Figure 3.* **The empirical tradeoff**: A visualization of results from Table 7 in SimpleQA Verified (Haas et al., 2025). Color indicates refusal rate. Most frontier models hug the diagonal (low refusal), optimizing for coverage at the cost of high hallucination rates. Achieving higher factuality requires aggressive abstention, imposing a severe utility tax (moving left). The unpopulated region near the ideal (gold star) illustrates the discriminative gap: current models lack the internal separability required to maximize factuality without destroying utility.

swerable and unanswerable questions (Zhang et al., 2025a). By incentivizing extended chain-of-thought and persistence, these models essentially prioritize the completion of a reasoning path over abstention, effectively rationalizing incorrect answers to satisfy the utility objective.

### 3.4. The Empirical Tradeoff

Lastly, we visualize the culmination of these factors in Figure 3, which plots the performance of various state-of-the-art

models on the SimpleQA Verified dataset from Haas et al. (2025). The plot reveals that the field has effectively bifurcated. Most frontier models (blue circles) hug the diagonal, optimizing for coverage at the cost of high hallucination rates. Conversely, models that attempt to maximize factuality (red circles) are forced to move left rather than up, paying the utility tax by discarding valid answers. The region of "ideal" performance—the top-right corner—remains entirely unpopulated. This empty space visualizes the discriminative gap: it is the region that we conjecture LLMs cannot currently reach, because they lack the intrinsic capability to distinguish their own hallucinations from their knowledge.

# 4. Faithful Uncertainty

In light of the challenges described in §3, we propose a pragmatic adjustment to the prevailing research objective. Rather than focusing solely on expanding knowledge, models should be as knowledgeable as possible *while faithfully expressing whatever uncertainty remains*. This does not abandon the factuality objective, as knowledge expansion remains valuable where headroom exists. However, it complements it with an objective that addresses those remaining cases where knowledge falls short and reliability is difficult due to the discrimination gap.

## 4.1. Defining the Objective

We adopt the framework proposed by Yona et al. (2024), defining faithful uncertainty as the alignment between the model's internal state and its verbalized output. We provide a self-contained overview of the main definitions and metrics in Appendix B. Specifically, faithful uncertainty requires an alignment between *intrinsic uncertainty*, i.e., the model's statistical confidence in the semantic meaning of its assertion (where high uncertainty implies a high probability of generating conflicting answers), and *linguistic uncertainty*, defined as the confidence expressed with words in the model's generated response, e.g., using phrases like "I am 90% sure," or "I might be mistaken". A model is said to *faithfully express its uncertainty* if its linguistic uncertainty accurately reflects its intrinsic uncertainty. Unlike fully eliminating hallucinations, which requires the model's output to match the *external world*, faithful uncertainty requires the model's output to match its *internal state*.

## 4.2. The Feasibility Argument

Faithful uncertainty bypasses the limitations discussed in §3. While mapping finite parameters to an infinite world is theoretically limited (Xu et al., 2024), mapping internal parameters to an output string is a fully observable, closed-loop problem. Even if there is no universal "truth direction" in the activation space that allows for perfect discrimination, the confidence signal is inherently computable from the

model's weights. The model does not need to know if the probability $P(\text{answer}) = 0.6$ corresponds to "truth" in the real world; it only needs to detect that its internal confidence is $0.6$ and map that signal to a verbalized hedge. Because the ground truth for faithfulness is internal to the system, it is theoretically solvable through architectural improvements, data modifications and better training recipes.

## 4.3. Reliable Utility

The faithful uncertainty objective directly addresses the utility tax (§3.2). Consider a set of answers where the model has 60% intrinsic confidence. If the confidence is well-calibrated, exactly 60% of these answers are correct. Under the goal of fully eliminating hallucinations, the model must make a collective decision for this entire set: to avoid the 40% of hallucinations, it must abstain on the whole cluster, thereby discarding the 60% of correct answers and hindering utility. Conversely, under the faithful uncertainty paradigm, the model preserves this utility: It generates the answers, but wraps them in appropriate epistemic markers. In this framework, a confident error remains an hallucination (albeit a faithful one), but an error wrapped in appropriate uncertainty is transformed into a useful hypothesis.

We define this outcome as *reliable utility*: the ability to maximize the volume of provided information without compromising user trust, achieved by aligning the decisiveness in which a claim is conveyed to the model's intrinsic confidence in it. Reliable utility mimics the way trust is established in human professionals. For example, we value a doctor not because they are all-knowing (omniscient), but because they faithfully communicate the distinction between a diagnosis they are certain of and a hypothesis they are merely testing. By allowing the model to answer when confident and hedge when uncertain, we can make models more trustworthy (Kim et al., 2024; Zhou et al., 2024) without making them less practically useful.

## 4.4. The Research Opportunity: Tractable Headroom

While this objective is theoretically feasible, it currently remains an unbridged gap. Yona et al. (2024) have demonstrated that current state-of-the-art models are far from satisfying this desideratum; they typically express high linguistic confidence even when their intrinsic uncertainty is low. Recent work has already begun to explore this capability, showing promising results using methods ranging from meta-cognitive prompting strategies (Liu et al., 2025), supervised fine-tuning (Eikema et al., 2025), and steering based on internal representations (Ji et al., 2025). Together with continued efforts to expand model knowledge, faithful uncertainty offers a path toward models that are both more knowledgeable and more trustworthy. We detail concrete challenges for advancing this direction in §6.

## 5. Metacognition in the Age of Agents

We have argued that fully eliminating hallucinations requires strong discrimination—the ability to separate what models know from what they don't—and that this is fundamentally difficult. The dominant strategy of *agentic AI* (Yao et al., 2022; Wang et al., 2024) might seem to sidestep this problem: with access to external tools, a model can in principle look up any fact. *Why know what you don't know if you can simply search?* We argue the opposite: tools do not remove the need for faithful uncertainty but amplify it. Without awareness of its own uncertainty, a model cannot determine when to invoke a tool (leading to inefficient overuse or dangerous under-use), nor can it appropriately weigh retrieved information against its own beliefs when conflicts arise. Faithful uncertainty thus becomes the control layer that governs tool use.

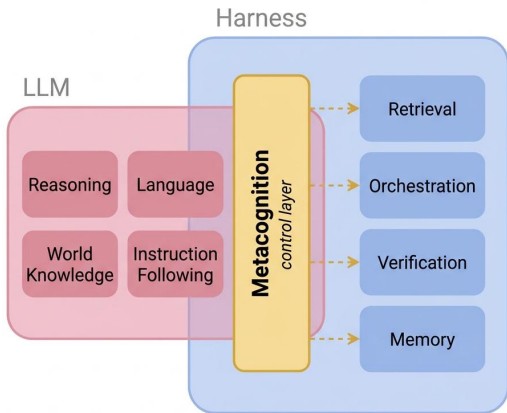

*Figure 4.* **A model that knows what it doesn't know can make the harness smarter and simpler.** The model's metacognition acts as the API or *control layer* (yellow) between the underlying LLM (pink) and the harness (blue). Without it, the harness is "flying blind" – it has to make all grounding and routing decisions externally, based on query type heuristics. With it, the agent can dynamically regulate behavior, e.g. retrieve only when confidence is low (efficiency) and express doubt when retrieved evidence conflicts with internal priors (reliability).

**Tool-Use Masks the Reliability Problem.** Current evaluations obscure this need. By focusing on final output accuracy, benchmarks reward successful retrieval without testing whether the model understood why it needed to search. Low hallucination rates may reflect retrieval quality rather than metacognitive competence—creating systems that are factually correct but unaware of their own limitations. This fragility is exposed when tools fail or return unexpected results; indeed, recent work (Qian et al., 2025; Lin et al.,

2025; Yan et al., 2026; Xu et al., 2026) shows that modern search agents lack such self-awareness, leading to systematic overuse.

**Storage vs Control.** Tools solve what we call the storage problem: the model need not encode every fact. But they introduce a control problem: governing the process of retrieval, verification, and orchestration — functions collectively managed by the *agent harness*, the scaffold that processes inputs, routes tool calls, and returns results. An agent must judge when its internal knowledge suffices and when to delegate to the harness – a decision defined by its uncertainty. When retrieval returns conflicting or low-quality information, the agent must weigh these signals against internal priors rather than blindly accepting whatever appears in context (Petroni et al., 2020; Li et al., 2023a). As illustrated in Figure 4, faithful uncertainty underlies all such control decisions.

**Towards Metacognitive LLMs.** Drawing on human metacognition (James, 1890; Son & Schwartz, 2002), we emphasize two processes: *introspection* (assessing one's own uncertainty) and *regulation* (adjusting behavior based on that assessment). Contemporary agents often rely on static heuristics or over-engineered harnesses. Future agents in open-ended environments require dynamic control: determining when information suffices, when to verify, when to halt. Instilling metacognition is thus not only a complement to eliminating hallucinations, but a prerequisite for reliable autonomous agents.

## 6. Call to Action

In this section, we offer concrete recommendations for the research community. We divide these into two categories: an overview of the main challenges and open problems for those exploring our suggested metacognitive LLMs and faithful uncertainty objectives, and practical suggestions for research on direct hallucination mitigation.

### 6.1. Challenges for Metacognitive LLMs

Unlocking models that can faithfully reflect their uncertainty requires solving several unique methodological hurdles:

**The Bootstrapping Paradox.** Base models, trained on authoritative internet text, rarely express doubt naturally (Yona et al., 2024; Zhou et al., 2024). Teaching the syntax of hedging (e.g., *"I am not entirely sure..."*) therefore requires supervised fine-tuning (SFT), but this creates a paradox: SFT datasets are static, whereas the "correct" uncertainty label is dynamic relative to the model's current state. Training on a static label of *"I don't know"* for a fact the model happens to know (or vice versa) induces hallucinated uncertainty or confidence. This requires developing infrastructure to support such dynamic datasets or methods to bootstrap

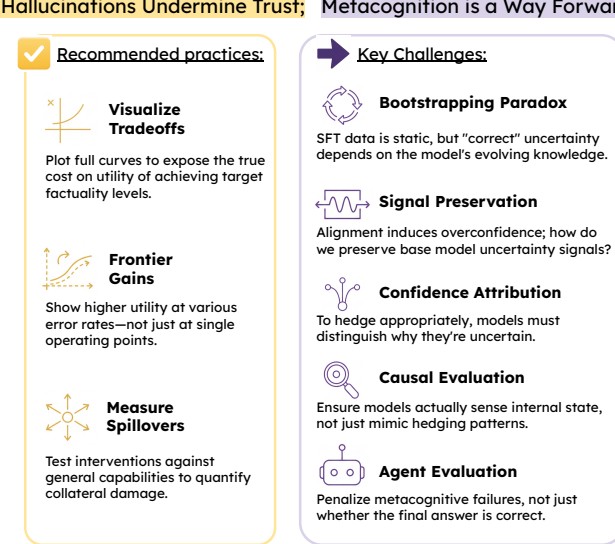

*Figure 5.* Recommendations for the research community.

the behavior of uncertainty (via SFT) without overfitting to possibly stale knowledge boundaries.

**Preserving the Signal through Post-Training.** Growing evidence suggests pre-trained models possess well-calibrated uncertainty representations, that are degraded during post-training (Kadavath et al., 2022; Achiam et al., 2023; Tian et al., 2023b; Zhu et al., 2023). Standard alignment techniques tend to induce mode-seeking behavior, making aligned models significantly more overconfident than their base counterparts (He et al., 2025; Song et al., 2025b). If our goal is to be faithful to the model's true knowledge boundary (often best captured by the base model), we face a conflict: how do we align models for safety and instruction following without erasing the subtle distributional information required for instilling metacognition? Developing such "uncertainty preserving" alignment algorithms is therefore an important direction for future work.

**Linguistic Precision Requires Confidence Attribution.** To effectively communicate uncertainty, a single scalar confidence score is insufficient (Delacroix et al., 2025), as a model may be uncertain for widely different reasons: ambiguity in the prompt (aleatoric), lack of knowledge (epistemic), ambiguity regarding the alignment behavior (normative), etc. High-quality faithful expressions of uncertainty therefore require the ability to trace the source of the uncertainty and map it to the appropriate linguistic hedge (e.g., *"It depends on what you meant by X"* vs. *"I do not recall X"*). Research should focus on disentangling these sources to yield a more informative and actionable signal.

**Rigorous Causal Evaluation.** A fundamental scientific risk is that models may learn to mimic the *style* of faithful uncer-

tainty without the substance, by learning simple heuristics (e.g., "always hedge when the prompt contains rare entities") rather than actually sensing their internal state. This is a general challenge in evaluating metacognitive abilities in LLMs, with recent works proposing approaches such as concept injection (Lindsey, 2025), cross-model evaluations (Kapoor et al., 2024; Binder et al.; Li et al., 2025a; Song et al., 2025a) and strategic games in which the model benefits from assessing and utilizing its own confidence (Ackerman, 2026). Developing proper evaluation frameworks is therefore a fundamental aspect of the overall research effort.

**Evaluating Metacognition in Agents.** Finally, as we move to agents, evaluation must shift from end-to-end correctness to process-based control. Standard benchmarks often reward agents that "luck into" the right answer despite bad reasoning (Yona et al., 2025). Agent evaluations could benefit from *control-centric* and model-dependent evaluations that seek to penalize metacognitive failures, such as searching for known facts (inefficiency) or trusting sources that conflict with known knowledge (sycophancy), regardless of whether the final answer happens to be correct.

### 6.2. Better Hallucination Mitigation Evaluations

The key challenge we identify is that a utility tax for eliminating hallucinations is unavoidable, and current evaluation paradigms are insufficient to fully capture and reflect it. Thus, for researchers working on methods aiming to fully eliminate hallucinations, we recommend three concrete evaluation practices:

**Visualize the Utility-Error Trade-off.** We recommend moving away from calibration-based metrics (ECE), which mask the discriminative gap by averaging over the whole distribution, but also away from summary discrimination metrics (like AUROC), which obscure specific operating costs. As we argued in §3, a model can be well-calibrated or have a decent AUROC yet still require prohibitive abstention rates to achieve high reliability. Instead, researchers should visualize the full Utility-Error Curve, as shown in Figure 2 (Right), which explicitly exposes how much utility must be sacrificed to achieve a specific target error rate.

**Demonstrate Frontier Improvements.** Claims of "reducing hallucinations" often amount to simply sliding along the existing tradeoff curve (e.g. increasing the refusal threshold) rather than improving the model's underlying capability. We urge the community to reject comparisons based on single operating points (e.g., "we achieved 95% accuracy"); Instead, contributions should demonstrate that for a fixed error rate, the method yields higher utility than the baseline.

**Measure Holistic Spillovers.** Finally, interventions must be tested for "collateral damage." Tuning a model to refuse long-tail queries often causes it to become evasive on "head"

knowledge or less helpful in reasoning coding or creative tasks. We recommend evaluating against a suite of such tasks to fully quantify the "cost" of an intervention as the degradation of helpfulness across the model's general capabilities, not just the lost recall on the target set.

# 7. Alternative Views

## 7.1. We Should Not Be Deprioritizing Factuality

A skeptic might argue that by emphasizing faithful uncertainty, we risk diverting attention from the important work of making models more knowledgeable. If resources shift toward metacognition, does progress on factuality slow?

We stress that faithful uncertainty is not a replacement for knowledge expansion, but a complement to it. Model providers already have strong incentives to develop models that are as knowledgeable as possible, and these efforts should continue. Moreover, different domains offer different headroom: in emerging areas like multimodal understanding, there remains substantial room for basic factuality improvements before the discrimination gap becomes the limiting factor (Cheng et al., 2025). Our proposal addresses cases where knowledge expansion alone falls short – when models encounter questions at the edge of their competence, faithful uncertainty ensures they communicate limitations rather than confabulate. The two objectives are synergistic: a more knowledgeable model with good metacognition is strictly better than either capability alone.

## 7.2. Users Prefer Confidence Over Uncertainty

From a product perspective, one might argue that users prefer decisive answers, and that this is partly why RLHF-aligned models are decisive. Constant hedging creates friction and may be perceived as incompetence, especially in creative or high-velocity tasks like coding and writing.

We stress that this objection ignores the specific scope of our argument (§2): faithful uncertainty, by design, does not target creative domains in which hallucination is indeed desirable. Furthermore, in long-form generation, faithful uncertainty need not be intrusive, and "localized" expressions of uncertainty, such as flagging a specific line of code or a specific date, can add value without blocking the user.

## 7.3. Latent Truth Exists, We Just Need Better Probes

A strong counter-position is that the challenges we describe in §3 are overstated. Proponents could argue that LLMs are incentivized to encode representations of truth (Ravfogel et al., 2025; Marks & Tegmark), so the main bottleneck may lie with current methods not yet sophisticated enough to extract it (Liu et al., 2023; Orgad et al., 2025; Gekhman et al., 2025). From this perspective, the pursuit of better discrimination should continue rather than be deprioritized.

We view the search for latent truth as a valuable pursuit that may well ease the tradeoffs we describe. However, it requires the strong assumption that a universal truth representation exists for the entire long tail of facts—an assumption we are skeptical of given the evidence in §3.2. Faithful uncertainty, by contrast, offers concrete headroom today. Unlike the "truth direction," we have robust evidence that models already possess accessible confidence signals that can be exploited. First, recent work in mechanistic interpretability demonstrates the feasibility of distilling self-awareness and confidence directly from the model (Stolfo et al., 2024). Second, reasoning models have been shown to be significantly better at expressing their confidence (Yoon et al., 2025; Podolak & Verma, 2025)—even while hallucinating more—suggesting that the metacognitive signal is distinct from the factual one. Finally, intrinsic signals (Chentanez et al., 2004) are already used as rewards in RL to encourage diversity (Kayal et al., 2025; Sukhija et al., 2025) and improve reasoning (Prabhudesai et al., 2025; Wang et al., 2025a; Li et al., 2025b).

# 8. Discussion

We have argued that fully eliminating hallucinations faces fundamental challenges due to a discrimination gap, and proposed faithful uncertainty as a complementary objective. This metacognitive awareness becomes increasingly important as LLMs evolve into agentic systems, where it serves as the control layer for robust tool use.

Faithful uncertainty connects to broader objectives in AI safety: at its core, it is a form of honesty—requiring models to accurately represent their epistemic state rather than project false confidence. Crucially, uncertainty communication enables appropriate human oversight, inviting users to verify, seek additional sources, or exercise their own judgment when models express doubt. Realizing this vision requires a shift in both model development (as current benchmarks focus on factual accuracy) and user expectations (users that expect LLMs to express their uncertainty, and can interpret that uncertainty appropriately).

# Acknowledgments

We thank Jonathan Berant, Alain Vaucher and Nitay Calderon for providing helpful comments on earlier drafts of this work.

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

## A. Additional Details

**Figure 2 Simulation Methodology.**   We created a synthetic dataset ($N = 25,000$) designed to reproduce the empirical confidence profiles reported by Nakkiran et al. (2025). We fixed a base hallucination rate of 25%. Confidence scores for correct answers ($y = 1$) and incorrect answers ($y = 0$) were sampled from Beta distributions, Beta($\alpha, \beta$), chosen to model overlapping confidence profiles typical of modern LLMs. Specifically, we sampled correct scores from Beta($1.8, 1.0$) (skewed toward high confidence) and incorrect scores from Beta($1.0, 1.3$) (skewed toward low confidence). To isolate discriminative power as the limiting factor, we applied Isotonic Regression to the raw scores. This enforced near-perfect calibration (smECE $\approx 0.014$), ensuring that the observed utility-error trade-offs stem purely from the overlapping distributions rather than miscalibrated probabilities. The utility-error trade-off curve was computed by sweeping a rejection threshold $\tau \in [0, 1]$ across the calibrated scores, calculating the proportion of total samples that were answered correctly (Utility) versus answered incorrectly (Error Rate) at each threshold.

## B. Faithful Uncertainty: Definitions and Measurement

We provide a concise overview of the key concepts necessary to operationalize the notion of faithful uncertainty (§4), drawing on Yona et al. (2024).

**Intrinsic uncertainty.**   We quantify uncertainty via the likelihood of generating conflicting answers under repeated sampling. Specifically, given a fact-seeking query $Q$ (e.g., "When was Barack Obama born?") and a candidate assertion $A$ (e.g. "1961"), if the assertions $A_1, \dots, A_k$ generated by $M$ under repeated sampling do not contradict $A$ (e.g., "1961", "I think he was born in 1961", or "August 4, 1961."), then we say the intrinsic confidence is high of $M$ in $A$ is high:

$$\mathrm{conf}_M(A) \;\equiv\; 1 - \tfrac{1}{k} \sum_{i=1}^{k} \mathbf{1}[A \text{ contradicts } A_i] \tag{1}$$

**Linguistic uncertainty.**   We quantify linguistic uncertainty via decisiveness, which reflects how confidently an assertion is conveyed through hedges, qualifiers, and epistemic markers. Given an assertion $A$ in a model response $R$, its perceived decisiveness is captured as the probability a reader would assign to $A$ being true based solely on the language of $R$ (and is implemented using LLM-as-a-judge, see (Yona et al., 2024; Liu et al., 2025)):

$$\mathrm{dec}(A;\, R, Q) \;=\; \Pr[A \text{ is True} \mid R, Q], \tag{2}$$

**Faithful uncertainty.**   A response $R$ *faithfully expresses* $M$'s uncertainty if its decisiveness tracks its intrinsic confidence assertion-by-assertion:

$$\mathrm{faithfulness}_M(R;\, Q) \;\equiv\; 1 - \frac{1}{|A(R)|} \sum_{A \in A(R)} \big| \mathrm{dec}(A;\, R, Q) - \mathrm{conf}_M(A) \big|. \tag{3}$$

A score of 1 indicates perfect alignment; lower scores reflect systematic over- or under-hedging relative to actual internal confidence.

**cMFG metric.**   Raw faithfulness scores are confounded by a model's confidence distribution. Yona et al. (2024) introduce *conditional mean faithful generation* (cMFG): expected faithfulness uniformly averaged across confidence levels (in practice, via equal-width confidence bins). A cMFG of 0.5 corresponds to a strategy whose decisiveness is independent of actual confidence. Current state-of-the-art models typically score 0.5–0.7, indicating that expressed uncertainty is only weakly aligned with intrinsic confidence.

