# OpenReview forum: "Position: Hallucinations Undermine Trust; Metacognition is a Way Forward"
_ICML.cc/2026/Position_Paper_Track — ICML 2026 Position Paper Track regular_

### Official Review · Reviewer_Qkjp · 2026-02-18

**Significance:** 3
**Argument Clarity:** 3
**Rating:** 5
**Confidence:** 4

**Questions:**

N/A

**Alternative Views Section:**

Yes

**Compliance With Llm Reviewing Policy A Conservative:**

Affirmed.

**Discussion Potential:**

4

**Final Justification:**

My rating is already positive. I will maintain my score.

**Paper Summary:**

This paper proposes that to engineer LLMs that are honest about what they do not know (rather than making up incorrect responses) we need training mechanisms that improve the models awareness of its knowledge boundary.
They define faithful uncertainty as the alignment between the model’s internal state and its verbalized output.

Assuming these metrics are tracked:
- accuracy (overall correctness)
- attempted accuracy (correctness on the subset for which an answer was attempted)
- calibration -- the alignment between confidence scores and empirical accuracy
- discrimination -- the ability to distinguish correct from incorrect answers based on confidence

The desideratum:
Rather than eliminating hallucinations by suppressing a massive volume of correct information, models should aim for maximal discrimination. Then, as long as the information about confidence is available to the model at all, this information will be accurately reported.


The models with high discrimination will only produce confident lies if there is no latent signal in the model at all that the answer may be wrong, and they will never suppress a possibly correct answer due to low confidence (as long as the confidence can be communicated).

The authors refer to discrimination engineering as a metacognitive problem.

They conclude with the list of open challenges, which future research should solve to implement this desideratum.
They recommend that benchmark performance reports should directly include the tradeoffs between attempted accuracy and response refusal.

**Position:**

Yes

**Position In Title:**

Yes

**Related Work:**

3

**Strengths And Weaknesses:**

Strengths:

Sound theoretical argument of why eliminating hallucinations is impossible.

The authors engage with alternative views.

Weakness:

Metacognition is a nice term, however in this context it is being overused. Metacognition has a rich meaning in psychology, beyond 'being able to report the internal confidence signal'.

There seem to be no real recommendation for *how* to implement discrimination, only a list of issues that will make it difficult to implement.

**Support:**

3

---

> ### Author Rebuttal · Authors · 2026-03-27
>
> We sincerely thank the reviewer for the careful reading and supportive assessment of our position paper ("sound theoretical argument of why eliminating hallucinations is impossible", "the authors engage with alternative views"). We now address the two main points flagged in the review.
>
> **W1: Overusing the term metacognition**: We agree that the term carries rich meaning beyond confidence reporting, and we attempted to ground our usage in that tradition: Section 5 draws explicitly from [1] and [2], distinguishing introspection (assessing one's own uncertainty) from regulation (adjusting behavior accordingly), and discussing their counterparts for establishing trustworthiness in LLMs. We do, however, consider an intentionally narrow scope (uncertainty about factual knowledge boundaries) and we will add a sentence clarifying this in the revision.
>
> **W2: No real recommendations for improving discrimination**: Indeed, the absence is intentional. We make the point (Section 3) that the community has been actively pursuing discrimination improvement for years through various means. Our position paper argues that these efforts are valuable but face a fundamental ceiling (“the discrimination gap”), so we intentionally shift the focus to the faithful uncertainty objective  - how to train models to honestly convey whatever uncertainty remains, rather than trying to close a gap we conjecture cannot be fully closed. The challenges we outline in Section 6 – the bootstrapping paradox, signal preservation through post-training, causal evaluation, agent-level metacognition – are specifically for this new faithful uncertainty objective. We fully agree they are fundamental and far from providing a full recipe; surfacing them as a novel research agenda and inviting the community to engage is one of the key objectives of our position paper.
>
> We welcome any additional suggestions for improving our work and thank the reviewer for their time and thoughtful feedback.
>
> References:
>
> [1] James, W., 1890. The principles of. Psychology, 2, p.94.
>
> [2] Son, L.K. and Schwartz, B.L., 2002. The relation between metacognitive monitoring and control. Applied metacognition, pp.15-38.

---

> > ### Author Rebuttal · Reviewer_Qkjp · 2026-04-01
> >
> > Thank you for the rebuttal. I will keep my score of 5.

---

### Official Review · Reviewer_R8TL · 2026-02-24

**Significance:** 2
**Argument Clarity:** 2
**Rating:** 3
**Confidence:** 4

**Questions:**

please refers to the weakness section

**Alternative Views Section:**

Yes

**Compliance With Llm Reviewing Policy A Conservative:**

Affirmed.

**Discussion Potential:**

3

**Final Justification:**

The response helped clarify the intended contribution of the paper as a conceptual reframing and research agenda around faithful uncertainty, and I agree this is a worthwhile direction. However, my core concerns remain only partially resolved: the main thesis is still under-formalized, the capability-vs-identifiability distinction remains important, and the empirical support is still somewhat limited for the scope of the claim. I therefore view the paper as interesting but still underdeveloped, and increase my score to 3.

**Paper Summary:**

This position paper argues that recent gains in factuality for large language models (LLMs) primarily reflect expansion of the knowledge boundary rather than improved awareness of that boundary. The authors suggest that eliminating hallucinations is fundamentally difficult due to limited discriminative ability, i.e., models can be calibrated on average but cannot reliably separate correct from incorrect answers instance-wise. They further argue that aggressive abstention imposes a “utility tax,” and propose “faithful uncertainty” as a more realistic objective. Extending the argument to agentic systems, they claim that uncertainty awareness becomes a necessary control signal for tool use.

**Position:**

Yes

**Position In Title:**

Yes

**Related Work:**

2

**Strengths And Weaknesses:**

### Strengths

1. Clear conceptual reframing: The distinction between calibration and discrimination is insightful and well-motivated. The argument that calibration does not imply instance-level error separability is technically sound and clarifies why simple thresholding strategies fail.

2. Correctly challenges an implicit community assumption: The paper convincingly argues that eliminating hallucinations is often treated as a straightforward engineering problem, whereas it may reflect structural limitations of likelihood-trained models.

3. Productive shift toward uncertainty-aware systems: The proposal to focus on faithful uncertainty rather than zero-error outputs is coherent and arguably more aligned with the probabilistic nature of language models.



### Weaknesses


1. Ambiguity between capability gap and identifiability limit: The central claim that models lack “strong ability to separate correct from incorrect answers (discrimination)” is rhetorically framed as a capability deficit. However, from a probabilistic modeling perspective, this may instead reflect an identifiability limitation under next-token likelihood training. If correctness labels are not directly supervised, instance-level truth discrimination is underdetermined in the long tail. In this interpretation, the discriminative gap is expected rather than surprising.

The paper would benefit from explicitly clarifying whether the limitation is: (A) representational (missing internal truth signal), or (B) statistical (insufficient supervision signal).


2. “Faithful uncertainty” reframes but does not solve hallucination: The proposal shifts the objective from correctness to calibrated epistemic behavior. While valuable, this does not eliminate incorrect outputs; it merely modifies how they are presented.

3. Limited discussion of alternative paths to discrimination: The argument assumes that discrimination cannot substantially improve without prohibitive abstention. What might be the alternative to this?

4. Lack of formalization: The discrimination argument could be strengthened with a formal treatment (e.g., separability conditions, ranking metrics, Bayes optimal decision boundaries). Currently, the claim is conceptually persuasive but not formally grounded.

**Support:**

2

---

> ### Author Rebuttal · Authors · 2026-03-27
>
> We thank the reviewer for the detailed feedback and for constructively engaging with our work. We are encouraged that the reviewer finds our core contributions to be valuable (calibration/discrimination distinction is "insightful and well-motivated," faithful uncertainty as "coherent and arguably more aligned with the probabilistic nature of language models”). We address the weaknesses below.
>
> **W1: The paper is ambiguous about whether limitations come from capability vs identifiability limits**. We appreciate this valuable distinction. We note, however, that the paper's core argument is robust to this debate: whether the discrimination gap reflects a representational capacity limitation or a statistical identifiability limit under next-token training, it exists in practice and produces the tradeoffs we discuss. We conjecture that a representational capacity limitation is the dominant factor in the regime the paper targets (long-tail factual knowledge). [1] show that linear truth encodings can emerge under next-token training, but under specific conditions that fail in our regime, where the training signal is too sparse to develop reliable discriminative representations. Importantly, even granting that the gap is primarily a supervision artifact, sparse coverage of long-tail facts means you cannot label what you cannot observe, making the two explanations functionally equivalent in this domain. We will add a paragraph in Section 3.2 clarifying this distinction, explaining our conjecture on which factor dominates in the long-tail regime, and why the argument holds under both interpretations. We thank the reviewer for pushing us to sharpen this.
>
> **W2: Faithful uncertainty reframes but does not solve hallucination**. We fully agree, and this is intentional. We argue that fully eliminating hallucinations is fundamentally difficult, so proposing yet another method to achieve zero errors would be inconsistent with our own diagnosis. The contribution of faithful uncertainty is not to eliminate incorrect outputs, but to make them honest rather than confident: an error wrapped in appropriate epistemic markers is far less harmful than a confident confabulation, and preserves the valid information that abstention would suppress. This reframing is precisely the point – we argue "reliable utility" (Section 4.3) is a better-defined and more tractable objective than zero hallucinations, and one that builds trust without destroying usefulness.
>
> **W3: Limited discussion of alternative paths to discrimination**: We address this directly in Section 7.3, where we engage with the counter-position that LLMs may already encode latent truth representations [1,2], and that better probing or training could substantially improve discrimination. We view this as a valuable and complementary research direction that we explicitly encourage. Our skepticism is specific: we conjecture that a universal truth representation for the entire long tail of facts is a strong assumption, and that the evidence in Section 3.2 weighs against it. Faithful uncertainty, by contrast, offers concrete headroom *today* without requiring that assumption to hold. We will make the connection to Section 7.3 more prominent in the revision.
>
> **W4: Claims are conceptually persuasive but not formally grounded**: As a position paper, we focused on synthesizing and reframing existing research rather than introducing new technical results. Specifically: (1) Figure 2 formalizes the utility-error tradeoff under explicit distributional assumptions. (2) Figure 3 grounds the discrimination gap empirically through results on frontier models on SimpleQA Verified. (3) For the impossibility arguments, we build on existing formal results [3,4]. We fully agree with the reviewer that formal treatment of the discrimination gap will strengthen the arguments made in the paper, and we view such attempts as natural and important next steps that this work invites.
>
> We welcome further discussion and again thank the reviewer for their time and thoughtful feedback. We believe that the revisions we committed to will substantially sharpen the paper's central argument. Noting the enthusiasm of the other reviewers for the paper's conceptual contributions, we hope these give reason to reconsider the current score.
>
> [1] Ravfogel, S., Yehudai, G., Linzen, T., Bruna, J. and Bietti, A., 2025. Emergence of linear truth encodings in language models.
>
> [2] Marks, S. and Tegmark, M., The Geometry of Truth: Emergent Linear Structure in Large Language Model Representations of True/False Datasets. In First Conference on Language Modeling.
>
> [3] Kalai, A.T. and Vempala, S.S., 2024, June. Calibrated language models must hallucinate.  56th Annual ACM Symposium on Theory of Computing
>
> [4] Kalavasis, A., Mehrotra, A. and Velegkas, G., 2025, June. On the limits of language generation: Trade-offs between hallucination and mode-collapse. 57th annual ACM Symposium on Theory of Computing

---

> > ### Author Rebuttal · Reviewer_R8TL · 2026-03-31
> >
> > Thank you for the rebuttal. The response helped clarify the intended contribution of the paper as a conceptual reframing and research agenda around faithful uncertainty, and I agree this is a worthwhile direction. However, my core concerns remain only partially resolved: the main thesis is still under-formalized, the capability-vs-identifiability distinction remains important, and the empirical support is still somewhat limited for the scope of the claim. I therefore view the paper as interesting but still underdeveloped, and update my score to 3.

---

### Official Review · Reviewer_Qaxt · 2026-03-07

**Significance:** 4
**Argument Clarity:** 4
**Rating:** 5
**Confidence:** 4

**Questions:**

1. How to masure the discrimination gap?  You cite Savage et al. (2025) and Taubenfeld et al. (2025) documenting weak calibration-discrimination correlation, but never compute AUROC for correct and incorrect separation on frontier models yourself. Do you have these numbers for the SimpleQA models in Figure 3?

2. Any metrics to measure the faithful uncertainty?  Would something like the correlation between verbalized confidence expressions and empirical accuracy be enough?

3. What are the limitations of devloping these kind of metacognitive agents for agentic coding tasks or other agentic applications?

**Alternative Views Section:**

Yes

**Compliance With Llm Reviewing Policy A Conservative:**

Affirmed.

**Discussion Potential:**

4

**Final Justification:**

I thank the authors for their rebuttal. They have addressed my concerns, so I lean towards accepting the paper, which is reflected in my score of 5.

**Paper Summary:**

This paper discusses a very important topic in the LLM literature, i.e., the hallucination problem in the current LLMs. The main position of the paper is that employing a metacognition strategy is the way forward. It introduces a difference between calibration and discrimination, highlighting that the discrimination in LLMs is the main bottleneck to address the hallucinations. I personally really like Figure 3 of the paper, which shows two diagonals and demonstrates that the top-right is empty due to the missing discriminative abilities in current LLMs. The paper proposes faithful uncertainty as an objective to train for to mitigate hallucinations. Overall, a good position paper.

**Position:**

Yes

**Position In Title:**

Yes

**Related Work:**

3

**Strengths And Weaknesses:**

Strengths:

1. Paper well written. The ideas of calibration vs. discrimination to understand the hallucination mitigation are clearly explained.

2. The objective of faithful uncertainty makes sense to balance the utility hallucination tradeoffs.

3. The challenges and future directions in developing metacognitive agents are clearly outlined.

Weaknesses:
1. Discrimination gap is more of a conjecture; no way is proposed to quantify or measure it.
2. No other original experiments are shown.

**Support:**

3

---

> ### Author Rebuttal · Authors · 2026-03-27
>
> We thank the reviewer for the positive assessment and the concrete questions. We are pleased that the calibration/discrimination distinction, the faithful uncertainty objective, and the challenges for metacognitive agents all came through clearly. We address the remaining points below.
>
> **W1 + Q1: Discrimination gap is more of a conjecture; no way to quantify or measure it**: If discrimination were perfect, AUROC (the area under the ROC curve for separating correct from incorrect answers using a confidence signal) would be 1.0, so any value below that directly quantifies the gap for the specific dataset. In response to the reviewer’s suggestion, we conducted a systematic review of AUROC evidence in the literature, which we will incorporate in the revision. This review reveals a consistent pattern across methods, models, and tasks: [1] report an average AUROC of 0.79 across 30 model×task combinations; [2] find GPT-4 tops out at 0.79 in medical QA; [3] find GPT-4o-mini reaches only 0.68–0.72 on biography generation – a long-tail knowledge domain similar to the one we study.  Crucially, *this 0.70-0.80 range is insufficient to escape the utility tax*. E.g., using the simulation framework from Figure 2, an AUROC of 0.79 still corresponds to a utility tax of ~40% when targeting a 5% error rate (at the 0.85 ceiling, the tax remains ~28%). While we do not have AUROC numbers specifically for SimpleQA, we find that the broader literature pattern strongly supports the discrimination gap conjecture.
>
> **W2: No other original experiments are shown**. Our goal was to diagnose the state of the field and propose a way forward –  a contribution we believe can be made convincingly through careful synthesis of existing evidence, without requiring new experiments. We note that per the ICML 2026 position paper CFP, position papers are expected to support their claims with “clear reasoning and evidence where appropriate”, and we believe our central claims are well-supported by the combination of original elements (Figure 2 simulation, Figure 3 empirical analysis) and the literature evidence we accumulate throughout. The one concrete empirical gap the reviewer correctly identified, AUROC for frontier models, is directly addressed above and will be incorporated in the revision. We would gladly welcome additional suggestions and are happy to conduct new experiments if the reviewer believes they would further strengthen the paper.
>
> **Q2: Metrics to measure faithful uncertainty?** [4] operationalize faithful uncertainty as the alignment between intrinsic and linguistic uncertainty and propose concrete metrics (cMFG), which subsequent work uses [5,6]. We thank the reviewer for identifying this gap, and in the revision, we will add an appendix section detailing the concrete definitions and metrics from prior work that are relevant to our discussion. We also note that as we discuss in Section 6,  there are still fundamental evaluation challenges: a model may produce appropriate hedging language without actually sensing its internal uncertainty. Distinguishing genuine metacognitive awareness from learned hedging patterns is an open evaluation problem we flag as a key challenge.
>
> **Q3: What about agentic coding tasks or other agentic applications?** Great point. Coding is a domain where execution provides a partial verification signal, reducing the discrimination problem, but there are still knowledge-heavy aspects where the type of metacognition we discuss matters: library versions, API compatibility, deprecated behavior. More broadly, for open-ended specifications where no executable ground truth exists, faithful uncertainty becomes important – a model that silently makes assumptions about ambiguous requirements is exactly the failure mode our framework targets.
>
> We hope our response addresses any remaining issues, and we welcome any further suggestions and discussion.
>
> [1] Farquhar, S., Kossen, J., Kuhn, L. and Gal, Y., 2024. Detecting hallucinations in large language models using semantic entropy. Nature
>
> [2] Savage, T., et al. (2025). Large language model uncertainty proxies: discrimination and calibration for medical diagnosis and treatment. *JAMIA*
>
> [3]  Becker, S., et al. (2025). Uncertainty Quantification for Hallucination Detection in Large Language Models: Foundations, Methodology, and Future Directions
>
> [4] Yona, G., Aharoni, R. and Geva, M., 2024, November. Can large language models faithfully express their intrinsic uncertainty in words?. In Proceedings of the 2024 Conference on Empirical Methods in Natural Language Processing
>
> [5] Liu, G.K.M., Yona, G., Caciularu, A., Szpektor, I., Rudner, T.G. and Cohan, A., 2025, November. Metafaith: Faithful natural language uncertainty expression in LLMs. In Proceedings of the 2025 Conference on Empirical Methods in Natural Language Processing
>
> [6] Eikema, B., Ilia, E., de Souza, J.G., Zerva, C. and Aziz, W., 2025. Teaching Language Models to Faithfully Express their Uncertainty

---

> > ### Author Rebuttal · Reviewer_Qaxt · 2026-03-31
> >
> > I thank the authors for their response. The authors have addressed my concerns. Therefore, I will keep my score of 5.

---

### Official Review · Reviewer_nAD3 · 2026-03-09

**Significance:** 4
**Argument Clarity:** 4
**Rating:** 5
**Confidence:** 4

**Questions:**

1. Do the authors have direct empirical measurements comparing discrimination metrics (e.g., AUROC on correctness prediction) across modern frontier models to support the claim of a fundamental discrimination gap?

2. How do the authors propose estimating intrinsic uncertainty in practice? Is it based on token probability distributions, internal representations, ensembles, or other signals?

3. What concrete metrics would determine whether a model is successfully expressing faithful uncertainty rather than merely mimicking hedging language?

**Alternative Views Section:**

Yes

**Compliance With Llm Reviewing Policy A Conservative:**

Affirmed.

**Discussion Potential:**

4

**Final Justification:**

Thank you for the detailed and thoughtful rebuttal. Overall, I support the acceptance of this paper, and my score already reflects this stance.

**Paper Summary:**

This paper argues that persistent hallucinations in large language models arise from a structural limitation: models often lack the discriminative capability to reliably distinguish between correct and incorrect outputs. The authors claim that most improvements in factuality to date have come from expanding the model’s knowledge boundary—for example through scaling, data expansion, or retrieval—rather than improving awareness of the boundary between known and unknown knowledge.

The paper proposes that this limitation creates a fundamental utility–factuality trade-off. Eliminating hallucinations requires aggressive abstention when uncertainty is present, which drastically reduces useful responses. Figures in the paper illustrate that reducing hallucination rates often requires discarding a large fraction of correct answers.

To address this limitation, the authors advocate a research shift toward faithful uncertainty, defined as alignment between a model’s internal confidence and its linguistic expression of that confidence. Under this framework, models should explicitly communicate uncertainty when confidence is low rather than producing confident but incorrect statements.

**Position:**

Yes

**Position In Title:**

Yes

**Related Work:**

4

**Strengths And Weaknesses:**

**Pros:**

1. The manuscript clearly articulates a concrete research position: metacognitive uncertainty modeling should become a central research priority for improving reliability in LLMs. The argument is consistent across sections and tied to concrete evaluation recommendations.

2. The utility–factuality trade-off is presented clearly and effectively. The figures illustrating the trade-off between hallucination reduction and answer coverage help formalize a phenomenon many practitioners observe informally. This framing could stimulate useful discussion in the community.

3. The paper follows a logical progression: empirical observations about hallucinations, theoretical limits, diagnosis (discrimination gap), proposed objective (faithful uncertainty), implications for agents, and recommendations for evaluation.This structure makes the position easy to follow.

**Cons:**

1. Limited empirical evidence supporting the central thesis. Most evidence is indirect or drawn from prior work.

2. The manuscript references theoretical work suggesting hallucinations are unavoidable, but the reasoning is only loosely connected to the proposed position. For example: Halting-problem style arguments about computability and Trade-offs between consistency and diversity. These results operate at a very abstract level and do not clearly imply practical limitations for real LLM systems.

3. “Faithful uncertainty” is an appealing concept but remains vague operationally.

**Support:**

3

---

> ### Author Rebuttal · Authors · 2026-03-27
>
> We thank the reviewer for the careful reading and supportive assessment. We address the weaknesses below.
>
> **W1 + Q1: Limited empirical evidence supporting the thesis.** We appreciate the push for more direct empirical grounding. In response, we conducted a systematic review of AUROC evidence in the literature, which we will incorporate in the revision. AUROC directly quantifies the discrimination gap: any value below 1.0 is evidence of imperfect discrimination. This review reveals a consistent pattern across methods, models, and tasks: [2] report an average AUROC of 0.79 across 30 model×task combinations; [8] find GPT-4 tops out at 0.79 in medical QA; [9] find GPT-4o-mini reaches only 0.68–0.72 on biography generation. Using the simulation framework from Figure 2, an AUROC of 0.79 still corresponds to a utility tax of ~40% when targeting a 5% error rate; even at the 0.85 ceiling, the tax remains ~28%. The tax only becomes negligible at AUROC ≥ 0.95 — well above anything reported in the literature. We believe the combination of evidence provides strong grounding for the central claims, but we are happy to conduct new experiments if the reviewer believes they would help.
>
> **W2: Theoretical arguments only loosely connected to the position**: We agree that the theoretical results we cite do not directly prove practical discrimination limits of the types we consider. Their intended role is to rule out the possibility of algorithmic “perfect fixes” in principle and to establish a *theoretical ceiling*, which motivates looking for empirical evidence of the *practical ceiling* (as we do in Sections 3.2 and 3.3); we will make this structure more explicit in the revision. We also agree that a more concrete formal treatment of the discrimination gap would strengthen the argument, and view this as an important direction this work invites.
>
> **W3 + Q2: "Faithful uncertainty" remains vague operationally and how to estimate intrinsic uncertainty**: [1], which we build on directly, operationalize intrinsic uncertainty as the probability distribution over semantically equivalent answers under repeated sampling, and define faithful uncertainty as alignment between this distribution and the model's verbalized confidence, yielding a concrete instance-level metric. Intrinsic uncertainty can also be estimated via semantic entropy [2], p(True) prompting [3], or internal representation probes [4], with different tradeoffs in cost and accessibility. In the revision, we will add an appendix section detailing the concrete definitions and metrics from prior work that are relevant to our discussion.
>
> **Q3: metrics beyond mimicking hedging:** The reviewer correctly identifies a deeper challenge: a model could achieve good cMFG scores simply by learning to hedge on questions that look hard (rare entities, unusual phrasings, etc) without *genuinely sensing its intrinsic confidence*. This is not a limitation specific to LLM - cognitive science research grappled with this for humans [5] and non-human animals [6,7]. Koriat [5] showed that human metacognitive accuracy largely reflects cue-based processing rather than privileged internal access – precisely the failure mode we are concerned about. This is one reason we situate the problem within a broader metacognition framework: it connects LLM evaluation to a tradition with methods for distinguishing genuine introspection from learned shortcuts (Section 6).
>
> We thank the reviewer for their time and thoughtful engagement.
>
> References:
>
> [1] Yona, G., Aharoni, R. and Geva, M., 2024, November. Can large language models faithfully express their intrinsic uncertainty in words? EMNLP 2024
>
> [2] Farquhar, S., Kossen, J., Kuhn, L. and Gal, Y., 2024. Detecting hallucinations in large language models using semantic entropy. Nature
>
> [3] Kadavath, S., Conerly, T., Askell, A., Henighan, T., Drain, D., Perez, E., Schiefer, N., Hatfield-Dodds, Z., DasSarma, N., Tran-Johnson, E. and Johnston, S., 2022. Language models (mostly) know what they know.
>
> [4] Orgad, H., Toker, M., Gekhman, Z., Reichart, R., Szpektor, I., Kotek, H. and Belinkov, Y., LLMs Know More Than They Show: On the Intrinsic Representation of LLM Hallucinations. ICLR 2025
>
> [5] Koriat, A., 1997. Monitoring one's own knowledge during study: A cue-utilization approach to judgments of learning. Journal of experimental psychology: General
>
> [6] Hampton, R.R., 2001. Rhesus monkeys know when they remember. Proceedings of the National Academy of Sciences
>
> [7] Smith, J.D., Schull, J., Strote, J., McGee, K., Egnor, R. and Erb, L., 1995. The uncertain response in the bottlenosed dolphin (Tursiops truncatus). Journal of Experimental Psychology: General
>
> [8] Savage, T., et al. (2025). Large language model uncertainty proxies: discrimination and calibration for medical diagnosis and treatment. JAMIA
>
> [9]  Becker, S., et al. (2025). Uncertainty Quantification for Hallucination Detection in Large Language Models: Foundations, Methodology, and Future Directions.

---

> > ### Author Rebuttal · Reviewer_nAD3 · 2026-04-02
> >
> > Thank you for the detailed and thoughtful rebuttal. Overall, I support the acceptance of this paper, and my score already reflects this stance.

---

### Decision · Program_Chairs · 2026-04-30

**Decision:**

Accept (regular)

**Comment:**

The paper takes the position that foundation models need to complement knowledge expansion with faithful uncertainty, i.e., honestly conveying whatever uncertainty remains. This metacognitive capability becomes even more critical for tool-augmented models, where it serves as the control layer that determines when to search and how to weigh conflicting information.

Overall, Three reviewers are quite happy and one review points out some downsides, such as main thesis rather under-formalized, the capability-vs-identifiability distinction remaining important, and the empirical support still being somewhat limited. Let me add to the downsides that it is not clear how this is different from the well-known argument of machines that know when they do not know, which advocates the use of (joint) distribution (probabilistic circuits) or at least scores, as well as uncertainty quantification. The latter has been used for LLMs, and there are also recent approaches on combining discrete diffusion models with probabilistic circuits. Additionally, there is neurosymbolic AI that also combines deep models with probabilistic models. The theoretical ceiling section should also mention Gödel's incompleteness theorem; it is clear that there cannot be an absolute truth machine. The metacognition could be extended, to avoid to just give a name to a vision, but the call to action is interesting. To add even more, while abstaining is an important part, it is not enough as we also need compositionality.

Overall, however, the majority is leaning towards acceptance, and the link to meta-cognition is interesting, but the authors should really provide a strong review of the literature.

https://ojs.aaai.org/index.php/AAAI/article/view/35094
https://www.nytimes.com/2023/03/27/science/ai-machine-learning-chatbots.html
https://proceedings.mlr.press/v235/cheng24i.html